# Uncovering the Damage Mechanism of Different Prefabricated Joint Inclinations in Deeply Buried Granite: Monitoring the Damage Process by Acoustic Emission and Assessing the Micro-Evolution by X-Ray CT

**DOI:** 10.3390/s25113332

**Published:** 2025-05-26

**Authors:** Wen Liu, Yingkang Yao, Yize Kang, Xiaojun Ma, Fuquan Ji, Ang Cao, Yuanyuan Wang, Nan Jiang

**Affiliations:** 1State Key Laboratory of Precision Blasting, Jianghan University, Wuhan 430056, China; lw13172975269@163.com (W.L.); kangyize@stu.jhun.edu.cn (Y.K.); m18943140597@163.com (X.M.); yuan2_wang@163.com (Y.W.); jiangnan@jhun.edu.cn (N.J.); 2Hubei Key Laboratory of Blasting Engineering, Jianghan University, Wuhan 430056, China; 3School of Civil Engineering, Sun Yat-sen University, Guangzhou 510275, China; 4CCCC Second Harbour Engineering Co., Ltd., Wuhan 430040, China; jifuquan_geo@163.com (F.J.); caoang0911@163.com (A.C.)

**Keywords:** buried project, pre-fractured granite, acoustic emission, RA-AF parameters, crack size function

## Abstract

This study reveals the damage mechanisms and fracture evolution characteristics of deeply buried granite with prefabricated joints (inclinations of 0°, 30°, 45°, 60°, and 90°) using uniaxial compression tests monitored by Acoustic Emission (AE) technology. Three-dimensional X-CT technology was used to analyze post-damage fracture evolution in specimens with varying joint inclinations. The results show that the stress–strain curve of deeply buried jointed granite under uniaxial compression includes three stages: initial compaction, crack extension, and failure. AE characteristics align with these stages, showing clear stress responses and timing features. In the initial compaction stage, micro-crack closure dominates, with smaller joint inclinations showing stronger closure effects. In the crack extension stage, joint inclination determines the crack propagation mode. In the failure stage, joint inclination significantly affects the spatial distribution of the rupture network by altering stress concentration areas and crack types. The proportion of shear micro-cracks increases with joint inclination, and peak strength rises with increasing joint angle, potentially accelerating micro-crack evolution. These findings provide valuable insights for designing excavation and instability monitoring in deeply buried multi-jointed granite underground projects.

## 1. Introduction

Owing to long-term geological processes, the rock mass commonly contains defects such as joints, cracks, pores, and structural surfaces. These features have a huge impact on the mechanical and physical characteristics of the rock. As a common deep rock [1], granite, under the influence of long-term geological stress and human interference (such as blasting, etc.), its geological structure becomes more and more intricate in deep engineering. Particularly as the depth of engineering burial increases, the burial depth of granite shows a positive correlation with self-gravitational stress. Over time, deep granite creates a tectonic stress field, and the combination of these factors places the deep granite formations under high stress. This intricate geological structure significantly influences the development and utilization of deep engineering projects. Deep granite is a type of rock whose original stress field is different from that of shallow rocks and is under the pressure of upland for a very long time, and it is produced in the granite by a variety of micro and macro-fractures (e.g., joints, fractures, and pores) [2,3]. These cracks, which are a typical type of discontinuous structural surface, significantly affect the strength and deformation characteristics of the rock mass. For instance, under upland pressure, the rock may display properties that transition from brittle to ductile to plastic. Such properties play a crucial role in determining the excavation response and support strategies in deep engineering projects.

In the last few years, many academics have performed a lot of experimental research by using prefabricated joints and have produced a large number of results [4,5,6] for the fracture damage behaviors and characteristics of jointed rock bodies. For example, Han et al. [7] produced soft and hard composite rocks with three parallel joints and conducted uniaxial compression experiments, which showed that the rock showed more obvious brittleness when the hard layer contained more joints; the change in peak strength with angle also depended on the rock layer where the cracks were located. The change in mechanical characteristics and percentage of tensile cracks were found to decrease and then increase as the amount of material increased, according to Liu et al. [8], who carried out uniaxial compression experiments on rock-like specimens with weakly filled rough joints. Feng et al. [9] found that cross-defected samples consistently showed “X”-shaped shear damage patterns, regardless of dynamic or static loading. Zhou et al. [10] studied crack growth in PMMA samples with two cross defects and classified the behavior into four crack initiation modes and five crack penetration modes. Chang et al. [11] analyzed the anisotropic strength and fracture mechanisms of rock specimens with cross joints, introducing concepts of multi-mode crack initiation and penetration. Li et al. [12] explored how preconditioning and existing single defects affect the damage behavior of marble under dynamic loading, finding that defects reduce dynamic strength and alter damage modes from tensile to shear-dominated. Wong and Einstein [13] and Bobet and Einstein [14] investigated the failure of marble with a single pre-existing defect under compression, describing it as a mix of initial tensile wing cracks and subsequent shear and tensile cracks at the defect tip. The static–dynamic characteristics of jointed rock bodies are the main topic of the above study, and it also looks at how jointed structural surfaces affect the mechanical characteristics of rock bodies more methodically.

Acoustic emission (AE) technology is an effective method for studying the entire process of internal fractures and damage evolution in rocks [15,16,17,18,19,20,21,22]. As acoustic emission technology continues to develop, an increasing number of scholars are using it to monitor the entire process of internal deformation and damage in rocks [23,24,25]. Du et al. [26] classified the rock failure process into five stages by using four types of rock specimens for uniaxial compression tests and AE monitoring, analyzed the crack development trend and damage characteristics of the rock using crack scales and crack demarcation lines, and found that the damage pattern of the rock could be accurately predicted by acoustic emission parameter analysis. Bi et al. [27] acquired mechanical and acoustic emission data throughout the rock damage and fracture process via uniaxial compression, Brazilian splitting, and modified shear tests on marble, granite, limestone, and sandstone. They determined the tensile and shear fracture boundaries of these rocks by correlating collected acoustic emission data (RA-AF values). Zhang et al. [28] analyzed the evolution of internal fractures in sandstone samples through uniaxial compression tests by examining acoustic emission (AE) characteristics under different water content conditions. (AE) demonstrated that the mineral composition and pore structure of sandstone could have an impact on the amplitude signals of the main frequency in acoustic emissions (AE). Zhang et al. [29] studied the acoustic emission behavior of limestone, sandstone, and rock salt under varying loading rates. Their results indicated that higher loading rates significantly restricted the development of acoustic emission activity in rocks. Moreover, Zhang et al. [30] observed that at low strain rates, the effect of strain rate on the amplitude and frequency trends of AE events was minimal. The findings of these studies demonstrate that acoustic emission technology is an effective tool for investigating the engineering safety of high-geostress rocks. However, existing studies have largely neglected the mechanical properties, macro-micro damage modes, and acoustic emission evolution characteristics of deeply buried hard rock prefabricated with different joint inclinations under uniaxial compression.

The paper is oriented to the safety construction needs of major projects in western China. Utilizing field specimens from a deep tunnel, a uniaxial compression test is conducted on artificially fabricated deep granite rock specimens featuring various joint inclinations; simultaneously, the impact of varying joint inclination angles on the damage accumulation process, damage modes, and tensile-shear crack propagation characteristics of granite subjected to high stress is examined using acoustic emission technology. Additionally, the critical point of rupture destabilization in jointed granite is determined based on the crack scale function, aiming to identify early warning indicators for granite damage under high-stress conditions. In addition, X-CT scanning technology is applied to reveal the internal fine-scale damage patterns of granite under different joint inclinations after the granite specimens are damaged. A theoretical foundation for the prediction of rock stability and the underground engineering prevention of disaster prevention can be obtained from the research conclusions of this thesis, which also provides a basis for further optimization of engineering design and safety assurance measures.

## 2. Sample Preparation and Experimental Methods

### 2.1. Granite Specimen Preparation

The samples were taken from a deep tunnel project in Western China, which is a typical high-stress hard rock area with a maximum depth of 1800 m, and the sampling interval was placed at a depth of 1600 to 1800 m. The rock sample is granite, characterized by a light grayish-white color and a blocky structure. It displays a fine to medium crystalline texture, and its mineral composition includes plagioclase feldspar, quartz, and a small quantity of mica. Rock specimens with joints were prepared in accordance with the recommendations of the International Society for Rock Mechanics (ISRM) [31], the standard cylindrical specimens of size of Φ50 mm × 100 mm were drilled along the normal direction of the primary joint surface using the core drilling method, the parallelism error between the two end surfaces of the specimens was controlled by the precision grinding process with the error of 0.1 mm, and the deviation of the diameter of the specimens was 0.05 mm. In order to characterize the fracture development under long-term high-geostress in the deep rock mass, the waterline cutting technique was used to prefabricate joints in intact specimens. The joint dip angles were cut at 0°, 30°, 45°, 60°, and 90°; the length of the joints is 25 mm and the width of the cut is 0.2 mm (see Figure 1). The accuracy of the notches was checked by laser confocal microscopy, the width deviation was controlled within ±0.02 mm, and the rock samples are shown in Figure 1.

### 2.2. Testing Systems

In this study, the RIST-416 multi-field coupled material mechanics test system was used to carry out uniaxial compression tests on prefabricated jointed granite specimens, which are equipped with a high-rigidity frame structure and a high-precision servo control system to ensure the loading stability under complex stress paths. The maximum axial load is 2000 kN, and the force measurement range is 20~2000 kN (full scale relative error ≤ ±1% FS).

Acoustic emission parameters, including ringing counts, energy, amplitude, and frequency distribution during rock deformation, were collected via real-time monitoring. Long-term geological processes often lead to defects such as joints, cracks, pores, and structural surfaces in rock masses, which significantly influence the physical and mechanical properties of rocks. The acoustic emission test utilized the Beijing Soft Island DS5-16B system for full-signal analysis, which captures micro-crack signals within the specimen during deformation. The system is equipped with six RS-2A sensors, with a preamplifier gain of 40 dB and a threshold of 45 mV. The critical temporal parameters consist of a peak detection latency (PDL) of 150 μs, an event identification time (HDT) of 300 μs, and an impact synchronization time (HLT) of 300 μs. Signals were sampled synchronously across multiple channels at 3 mHz to identify critical features of the acoustic emission signals.

Following the uniaxial loading test, each specimen underwent X-ray computed tomography (X-CT) to visualize the internal crack penetration within the damaged specimen. X-ray computed tomography (X-CT) is a non-invasive imaging modality that reconstructs localized density variations based on X-ray attenuation within the specimen. A nanoVoxel-4000 high-resolution micro-CT system was used to perform the CT scan, which was run at 220 kV source current of 180 A at a source voltage of 220 kV, and 1080 frames were scanned every single rotation. The scanning was conducted with a voxel size of 0.05 mm × 0.05 mm × 0.05 mm, resulting in an image with an isolation layer thickness of 0.117791 mm. From the CT images, the influence of joint inclination on crack consolidation and spalling is clearly observable.

### 2.3. Experimental Procedure

Uniaxial compression, acoustic emission, and post-test CT scan tests were performed on granite specimens with prefabricated joints of different inclinations. First, the pre-fractured specimens were subjected to uniaxial compression, and acoustic emission was monitored in the laboratory to record fracture information. At the end of the test, 3D CT scans showed crack penetration patterns in the rock bridge region of the fractured specimens. The test procedure involved displacement-controlled loading at a rate of 0.12 mm/min [32], in accordance with the ASTM D7012 standard [33]. A 2 kN preload was applied at 5 mm/min to ensure indenter-specimen contact, then loading continued at 0.12 mm/min until macroscopic damage occurred (Figure 2). (2) Longitudinal Wave Velocity Measurement: Using the lead break method, petroleum jelly was applied between the sensor and sample for proper coupling. During testing, movement was prohibited to maintain a quiet environment for accurate readings. (3) CT Slicing and 3D Reconstruction: 3D images were rebuilt from volumetric data using Avizo software (2022 version) for damaged specimens. Four CT slices parallel to defects were taken from the upper, middle, and lower regions at d = 30, 40, 50, and 60 mm. For intact granite (Figure 3), four CT slices sectioned the 3D model along the XY direction.

## 3. Result and Analysis

### 3.1. Stress–Strain Curve

The damage progression of the deep granite samples mostly consists of four stages: initial compaction–densification, linear elasticity, plastic yielding, and damage, according to the results of the uniaxial compression test of deeply buried granite specimens. Nonetheless, there are notable variations in the stress–strain curves of specimens with varying joint tilts as well as those of intact granite.

As illustrated in Figure 4, the intact granite specimens exhibited a gentle slope in the initial compaction stage, with an elastic modulus of 5.397 GPa, while the specimens at α = 0° (with joints oriented perpendicular to the loading direction) exhibit a markedly lower initial stiffness of 2.202 GPa, illustrating the combined effects of primary micro-crack closure and joint compression. Intact specimens keep a clear stress–strain linear relationship with peak elastic modulus in the linear elastic phase, meanwhile, the elastic modulus of specimens with a value of 0 decreases to 3.590 GPa. During the plastic yielding phase, the curve slope increases, and α = 0~60° specimens exhibit step-like stress fluctuations corresponding to crack bifurcation and propagation. In the post-peak damage stage, intact and α = 90° specimens undergo brittle fracture, whereas α ≤ 60° specimens retain residual strength due to shear-slip mechanisms.

### 3.2. Mechanical Parameters

The regulatory effect of the joint dip angle α on mechanical properties is substantial, with the specific mechanical parameters of the granite specimens presented in Table 1. The uniaxial compressive strength showed a nonlinear increase, ranging from 32.1 MPa at α = 0° to 136.4 MPa at α = 90°, with the intact specimen reaching 160.97 MPa. The modulus of elasticity demonstrated an exponential growth trend with α increase, with E = 2.202 GPa at α = 0°. The recovery of the stiffness was more than 92% of the intact specimen at α ≥ 60°, indicating that the closure effect of the joint surface dominated the recovery process. The damage chronological analysis demonstrated that the post-peak damage time was at its maximum at α = 45°, which was 65.3% longer than at α = 0°, during the interval of maximum energy accumulation rate. The damage mode evolves with the angle α: specimens with α ≤ 45° show a sawtooth stress drop dominated by shear slip, while specimens with α ≥ 60° transition to a tensile-shear composite brittle damage, which is microscopically manifested as a synergistic extension mechanism of nascent cracks and joint surfaces, as shown in Figure 5.

### 3.3. X-CT Micro-Damage Mechanism Analysis

The progression of the micro-cracks inside the rock is represented by damage to the rock material [34]. To better understand how different joint dip angles affect micro-crack formation and propagation in granite specimens, this study employs X-ray computed tomography (X-CT) for 3D reconstruction of damaged specimens and analyzes fine-scale fracture evolution in granite with varying joint dip angles. Typically, rock damage morphology is only visible to the naked eye on the specimen’s inner surface. While AE can detect micro-cracks at a microscopic level, its assessment of crack deformation characteristics is somewhat limited due to noise signals and positioning accuracy [35]. However, combining AE with X-CT integrates laboratory-scale deformations with microscopic injuries, creating complementary advantages. Thus, using both techniques together enables a more comprehensive exploration of the correlations between macroscopic surface deformation and microscopic internal damage.

Four tomographic slice sets (L = 30 mm, 40 mm, 50 mm, and 60 mm) were acquired at 10 mm intervals along the specimen’s longitudinal direction (Figure 6). In CT images, cracks appear as low-density regions due to differences in density. Crack sprouting typically starts at stress-concentrated crack tips. Figure 6 details crack locations for all test specimens. The crack distribution patterns vary among specimens subjected to compression at different joint angles. Analysis of the damage characteristics indicates that the location of crack sprouting is primarily determined by the joint dip angle α.

Specifically speaking, the intact granite specimen showed a typical shear-split composite damage pattern, with the primary crack running through the specimen at an inclination of 55 ± 5°, and secondary cracks distributed radially at the loaded end. The outer surface of the specimen was severely damaged, and more loose stones were generated during the loading process. When α = 0°, the primary crack propagates vertically along the prefabricated joint surface, forming an axial fracture zone through the specimen. Secondary cracks are distributed parallel to both sides of the joint surface, indicating that low-inclination joint surfaces induce stress concentration and suppress lateral crack bifurcation. In slices of L = 50 mm, the density of secondary cracks decreases with depth. When α = 30°, the primary crack still extends along the prefabricated joint surface, but secondary crack density increases by 30% compared to α = 0° specimens, with cracks bifurcating in an arc around the joint surface. When α = 45°, the primary cracks extend perpendicular to the tip of the joint surface. L = 40 mm and L = 50 mm slices show clusters of conjugate shear cracks forming an ‘X’-shaped network with other cracks, indicating that α = 45° jointing promotes multi-directional crack expansion. When α = 60°, the primary crack deviates from the prefabricated joint plane, deflecting 10~15° toward the loading axis, with secondary cracks forming a branching fractal network at the ends. L = 40 mm slices show multiple micro-cracks inside the cleavage-state specimen. When the main crack runs almost vertically (inclination 75° ± 5°), the damage characteristics of intact specimens are similar to the damage parameters of the specimen; however, starting from the upper tip of the vertical joint, it is not extended to the horizontal crack. L = 40 mm and L = 50 mm slices show macroscopic cracks within the specimen, with densities 30% lower than those of intact specimens, confirming weakened control of rupture paths by high-dip joints. As joint inclination increases, spalling on the specimen’s outer surface and the length, average width, and area of internal cracks also increase. Crack propagation paths extend from the joint tip, with specimen spalling occurring around the joint.

In order to quantitatively analyzes the crack sizes in the rock bridge segments of granite specimens and to examine the effect of pressure damage in the axial direction on the fracture of granite specimen segments with different joint inclinations, a series of digital image processing (e.g., binarization, edge detection, region growing algorithms, etc.) was carried out for the extraction of the cracks. Figure 7 depicts the size of the extracted cracks. From the CT volumetric data, the three-dimensional crack mesh pattern of the rock bridge of the granite specimen can be extracted. There is a good correspondence between the main cracks obtained from three-dimensional reconstruction and the macroscopic failure mode of the specimen. The specimen also contains numerous secondary cracks. CT imaging shows that shear cracks and induced damage occur in jointed granite specimens with α = 0°, with the crack count at rock bridges minimized. Intact specimens exhibit the largest crack size, which increases in jointed specimens with α. The size of cracks at α = 90° is comparable to the size of cracks in intact granite specimens. α = 30° to 60° has a relatively simple crack network, with shear cracks mainly at the ends of the joint region of the specimen.

There are marked distinctions in the damage characteristics between intact granite specimens and those with prefabricated joints. Stress concentration near prefabricated joints primarily dictates where cracks initiate, while the stress distribution around these joints determines where cracks first form. This crack formation is mainly governed by geometric factors rather than the loading rate [36]. As the joint dip angle α increases from 0° to 90°, the average dip angle of the main fracture vertically asymptotically changes from 90° to 75°, negatively correlated with the joint dip angle. This indicates that joint geometry controls crack propagation direction via stress deflection. According to engineering classification criteria for uniaxial compressive peak strain (ε_m_) [37], damage is brittle when ε_m_ < 1%, transitional when 1% < ε_m_ < 5%, and ductile when ε_m_ > 5%. In this test, the axial peak strain of the granite specimens was within 1%, reaching the critical value for brittle damage. Combined with the primary crack penetration characteristics and the cross-section characteristics of secondary cracks observed by CT, it can be seen that brittle shear-split composite damage is the main damage mechanism of the specimens.

### 3.4. Acoustic Emission Evolution Characteristics

#### 3.4.1. Acoustic Emission (AE) Signal Characteristics and the Spatial Distribution of Events

Acoustic emission technology is a well-established and precise non-destructive testing method for monitoring micro-fracture activity within rock samples. Analyzing five groups of uniaxial compression tests with varying joint inclinations (α = 0°, 30°, 45°, 60°, and 90°) along with intact granite, we examined the variations in acoustic emission parameters, including ringing counts, energy release rate, RA-AF values, and 3D localization points, to explore the relationships among these parameters throughout the fracture process, from micro-cracks to macro-fractures. The analysis of the synergistic response mechanism of acoustic emission parameters during the cracking process is shown in Figure 8.

Figure 9 demonstrates the evolution of ringing counts (RC), cumulative ringing counts (CRC), and locus energy release with stress versus time during uniaxial compression of granite specimens with different joint inclinations [38]. The three-dimensional localization technique of acoustic emission reveals the rupture evolution law of granite samples during uniaxial compression, and the spatial distribution of the localization points and the energy release characteristics can characterize the damage accumulation process of samples with different joint inclinations; combined with the stage division of the stress–strain curve, the acoustic emission activity shows a significant phase-by-phase response characteristic. During the uniaxial compression process, based on rock crack propagation patterns, acoustic emission ringing count trends, and energy localization point changes, this study categorizes the testing process into three phases: the initial compression phase, the crack propagation phase, and the failure phase.

During the initial compression stage, the axial load is low and the acoustic emission signal originates from primary micro-crack closure. Regarding intact granite specimen parts, the bottom of the specimen is very few in the acoustic emission events, indicating that the damage evolution is mostly dominated by end-primary micro-crack closure. On the other hand, different rupture evolutions are seen in prefabricated joint specimens. At α = 0~30°, the closure impact is stronger, reducing cumulative ringing counts, as the joint surface nearly parallels the loading axis. When the specimen has a specific inclination angle because of the joint surface’s orthogonal loading direction, the specimen is in a certain direction, and the main micro-crack closure is restricted, which greatly increases the number of rings.

Entering the crack extension stage, the primary crack closes and the specimen begins to produce shear-tension cracks. All specimens show a sharp increase in ringing counts and micro-cracks converge in a small area around the prefabricated joints, but there is significant divergence in the evolutionary pattern. Acoustic emission events in intact granite progressively shift toward the specimen’s center as stress increases, indicating progressive crack propagation from the ends to the center. When the α = 0°, the specimen’s ringing counts of the specimen increase stepwise. When α ranges from 30° to 45°, the joint surface is inclined to the loading direction, causing secondary cracks to branch and expand. This process is synchronized with the shear-slip behavior during the plastic yielding stage of the stress–strain curve and occurs in a continuous growth pattern. When α ≥ 60°, the rapid accumulation of acoustic emission (AE) events during the final stage of elastic deformation migrated toward the ends of the specimen, reflecting the characteristic of shear cracks propagating in an oriented manner along the direction of the maximum principal stress. It is noteworthy that the density of acoustic emission events is generally higher in the mid-end region than at the end of all the specimens, which is mainly attributed to the dominant distribution of micro-cracks induced by unloading damage from the processing of the rock samples and the non-uniform redistribution of the stress field during compression. The ringing count rate surges to a peak before declining. The specimens with α ≤ 60° exhibit an earlier decline phase, indicating that granite specimens with smaller joint dip angles are more prone to the formation of a larger number of micro-cracks.

During the injury stage, ringing counts increase at a slower rate. Intact granite specimens experienced the formation of a macroscopic shear surface, which caused through-cracks and specimen fragmentation, resulting in a sudden drop in the slope of the cumulative curve. For specimens with α ranging from 0° to 60°, the number of acoustic emission events increases due to complex crack propagation caused by shear slip. The location of these events shifts toward the ends of the specimens, with micro-cracks extending along the prefabricated joints at an angle. In contrast, specimens with α = 90° and intact specimens show a more gradual increase in ringer counts due to the homogenization of crack paths (main crack inclination of 75° to 90°). Some spalling occurs on the specimen surfaces, and acoustic emission events become relatively quiet. This indicates that the joint dip angle significantly influences the spatial configuration of the 3D fracture network by altering stress concentration zones and crack propagation types.

#### 3.4.2. AE Energy Characterization and Tensile-Shear Phase Damage Analysis

In this study, the synchronization of specimen cracking with an acoustic emission system effectively monitored the development of macro-micro cracks, overcoming inherent methodological limitations and revealing the damage and fracture processes during granite specimen uniaxial compression testing.

The energy and macro-micro crack evolution of the intact granite specimen at a loading rate of 0.12 mm/min are shown in Figure 10a. After loading for 422.6996 s, the closing stress reaches 16.959 MPa, and the specimen enters the stage of local crack compaction and closure. At this point, shear cracks slightly outnumber tensile cracks, while the acoustic emission energy remains relatively stable with minor fluctuations. After loading for 737.5 s, the crack initiation stress is 71.389 MPa, and cracks begin to form in the granite specimen. The percentage of tensile cracks decreases to 45.78%, indicating that internal damage dominates before significant macroscopic damage occurs, while shear cracks continue to propagate. After 1131.52 s of loading, the specimen reaches its maximum strength of 160.935 MPa and the acoustic emission energy increases exponentially, indicating strong energy release inside the granite specimen. The proportion of tensile cracks stabilizes, indicating that tensile cracks start to interact with shear cracks, eventually reaching equilibrium.

From a rock mechanics perspective, the crack development in granite specimens with different joint inclinations exhibits notable differences, as compressive and tensile stresses oscillate continuously with crack initiation and propagation.

The acoustic emission energy and the evolution of shear-tension cracks in granite specimens with varying joint dip angles are depicted in Figure 10b–f. From the figure, it is evident that the acoustic emission energy of jointed granite specimens over time closely mirrors the ringing count. In the initial to intermediate stages, both parameters maintain low levels. But when trying tests to see if there is a specimen failure, an obvious increase in the amount of acoustic emission activity is seen. The curve of ringing counts shows similar characteristics across the initial compression, crack propagation, and failure stages. This indicates that as the granite specimens approach failure, their acoustic emission activity intensifies, reflecting both a higher number of events and a greater implied energy release.

When α is between 30° and 45°, shear cracks slightly exceed tensile cracks in the early loading stage. Stress and strain distribution is relatively dispersed, leading to fewer concentrated cracks. As the angle increases, the ratio of shear cracks increases. When α is between 60° and 90°, shear cracks account for approximately 60%. During the elastic loading phase, cracks accumulate with loading energy, tensile cracks become less common, and strain concentrates around the joints and in conjugate regions centered on the joints. As loading proceeds to the destructive yield stage, the local deformation zone around the joint extends along the tip direction to the end of the specimen, forming a different fracture zone oriented with the joint tip direction. The specimens show a visible macroscopic main crack and exhibit mixed shear-tension damage along the joints, with shear damage being dominant.

#### 3.4.3. RA-AF Characteristics (RA) and the Ratio of Ring Counts to Duration (AF)

In JCMS-III B5706 (2003), a method was proposed to differentiate the types of cracks in concrete materials based on the amplitude (RA) and the ratio of ring counts to duration (AF) [39,40,41,42]. Tensile cracks are linked to acoustic emission (AE) events with smaller RA values and larger AF values, whereas shear cracks are related with AE events with larger RA values with smaller AF values, as shown in Figure 11. Building on this understanding, the distribution of RA-AF values has been analyzed to explore how crack propagation occurs in high-geostress granite under uniaxial compression with varying joint inclinations [43,44,45,46]. *AF* and *RA* calculations are outlined as follows.(1)$$AF=\frac{RC}{DT}$$(2)$$RA=\frac{RT}{A_{dB}}$$

In the context, *AEC* denotes ring counts, *DT* signifies duration time, *RT* indicates rise time, and *A*_d*B*_ represents the maximum amplitude. *AF* is measured in kHz, while *RA* is quantified in ms/V.

Under uniaxial compression conditions, the RA and AF values of the acoustic emission signals of high-geostress granite with different joint inclinations at the monitoring sites were determined. The results are presented as scatter plots showing the distribution of RA-AF values (see Figure 12). The AE data points are clearly concentrated where RA exceeds AF, indicating a higher incidence of shear cracks than tensile cracks. RA values for intact granite specimens subjected to high ground stresses and specimens with joint inclinations were mainly in the range 0~3.3 μs/mV, while AF values were mainly in the range 0~350 kHz. The proportion of shear cracks increased as the inclination of the prefabricated joints increased. The percentage of shear cracks in an intact specimen is 51.90%, increasing to 60.83% when the joint is inclined at 60° and decreasing to 53.29% when inclined at 90°. The percentage of shear cracks increases from 0° to 60° of joint inclination, after which it decreases as the inclination reaches 90°. Due to joint inclination, force transmission in the specimen is non-uniform, leading to shear crack formation. When joint inclination is greater, shear crack growth is somewhat limited due to the force direction being perpendicular to the inclined plane.

Figure 13 demonstrates how the ratio of tensile to shear cracks evolves during the damage process in intact granite and granite with varying joint inclinations. With an increase in α, the number of shear cracks initially rises and then declines. This shows that increasing the joint dip angle significantly affects crack type, leading to more shear damage during deformation. When 0° < α < 60°, the joints are more parallel to the vertical center of the granite, keeping the tensile crack percentage above 40%. However, as α increases, the jointed granite undergoes progressive deformation, allowing more micro-crack development and plastic deformation. At a loading rate of 0.12 mm/min, localized deformation in jointed granite accelerates, reducing the time for crack nucleation and plastic deformation. This results in a more brittle failure mode, with shear cracks becoming more widespread as the joint dip angle increases.

This phenomenon matches the X-CT micro-fracture pattern of granite specimens. More precisely, shear micro-cracks progress into the final macroscopic failure pattern, relying on the interplay and cumulative effects of tensile and shear cracks throughout the compressive loading process. However, as α increases, the failure effect intensifies, restricting the time for shear deformation to accumulate sufficient crack propagation energy. Consequently, there is somewhat less of a tensile crack in the share. As the response of jointed granite moves toward more extreme brittle failure, the proportion of shear cracks slightly decreases at α = 90°, nearing that of intact granite specimens, which are marked by a relatively higher number of tensile cracks.

#### 3.4.4. Evolution of Micro-Crack Scaling Using Acoustic Emission B-Values

The acoustic emission (AE) crack size function is a statistical indicator based on amplitude in acoustic emission monitoring. In parameter-based AE analysis, statistical processing of AE parameters is typically performed to derive relevant statistical indicators. In this study, a logarithmic transformation is applied to preprocess the amplitude data of AE signals to linearize the dynamic range of the data. The formula involves taking the logarithm of the original AE amplitude data *x*, converting it to log_10_(*x*), thereby linearizing the data, as follows:(3)$$\log_{x}=\log_{10}(x)$$

To analyze the frequency–magnitude distribution of the data, a sliding window is defined with a size of W and a step size of Δ*s*.(4)$$N=[\frac{L-W}{\Delta s}]+1$$
where *L* is the total length of the data. Within each window, we calculated the cumulative distribution of amplitudes and obtained log_10_(*N_c_*(*j*)) by logarithmic transformation, where *N_c_*(*j*) is the cumulative number of events in the *j*th bins and the cumulative count is(5)$$N_{c}(j)=\sum_{i=j}^{n}n_{j}$$

The logarithm of the resulting cumulative counts was used to determine the *b*-value using a linear fitting method, which describes the magnitude distribution of the event. The fitting equation is(6)$$\log_{10}(N_{c})=A-b^{\prime }\log_{10}(C)$$
where *C* is the distribution center and *b’* is the slope obtained from the fit. The *b*-value was determined as *b* = −*b′*.

The end index *t_k_* of each window is recorded and is used to mark the point in time at which the result of the calculation was made, as follows:(7)$$t_{k}=i_{k}+W-1$$

Figure 14 presents the stress versus b-value curves over time during the rupture of intact granite and granite with different joint inclinations under uniaxial compression. The *x*-axis represents the normalized loading time. To ensure accuracy, real-time b-values of acoustic emissions are calculated using a sliding window approach. The number of events within the window varies, typically from tens to hundreds, depending on the specimen’s deformation and failure phase. For intact specimens, the b-value curves are regular, and crack development is uniform during the initial compaction and crack propagation stages, fluctuating around b = 1.5. Upon reaching the peak strength and damage stage, the b-value curve steeply increases, corresponding to significant damage in the granite specimen, marked by the formation of large cracks.

In comparison, granite samples with pre-made joints showed considerable variability. During initial compression, specimens with α = 0~30° exhibited high fluctuating b-values that gradually rose with loading. For α = 45~90°, b-values showed minor fluctuations, indicating stable development of cracks across scales with little change in acoustic emission event proportions. In the plastic phase, b-values for α = 0~90° decreased, cracks developed slowly (marked by increasing low-energy acoustic emission events and decreasing high-energy ones), and small cracks grew gradually. During the damage phase, b-value changes mirrored the stress–strain curve, reflecting accumulating damage. As the load neared peak strength, small cracks merged and grew rapidly, indicating specimen damage. A sharp b-value drop signaled stress-induced damage, but due to pressure constraints, some cracks continued growing to form macroscopic fractures.

During uniaxial compression damage, acoustic emission events typically exhibited high energy, with events of varying energies occurring throughout the process. Analysis combining the stress–strain curves of granite specimens reveals that abrupt shifts in the b-value are strongly associated with rock stress drops, signaling brittle rock damage. When used as a monitoring parameter, the b-value permits more accurate capture of the entire rock damage process, from micro-crack initiation through growth to macro-fracture formation.

Figure 15 presents the b-value statistics of acoustic emission events at different stress levels during the specimen’s failure under uniaxial compressive load. From the figure, it is obvious that when the stress level is below 40%, only a few acoustic emission (AE) events are observed, and the AE b-value starts to decrease gradually. This happens because the fraction of small-scale micro-cracks begins to increase at lower stress levels. These micro-cracks generate low-energy low-amplitude AE signals, affecting event localization precision. Once the stress level exceeds 40%, the number of AE events that are localizable grows quickly, which is followed by clustering in certain areas and an increase in the AE b-value. This shows that the percentage of large-scale micro-cracks increases gradually. As the stress nears 80%, the b-value peaks, event clustering is prominent, and macroscopic rock cracking occurs. When the stress level exceeds 80%, cracks propagate within the rock, ultimately causing instability and damage.

The fracture evolution of jointed granite was analyzed using real-time acoustic emission monitoring and post-test X-CT scanning to study the macro-fine-scale fracture behavior of granite specimens with prefabricated joints. Acoustic emission monitoring and X-CT fine-scale scanning techniques were employed to incorporate internal and external surfaces and macro-fine-scale reference factors for jointed rock damage. In this study, real-time acoustic emission monitoring under uniaxial compression conditions revealed abrupt changes consistent with the stress–strain curve. Additionally, a synergistic analysis of acoustic emission spatial localization and X-CT micro-crack distribution detection was conducted. These approaches helped address the challenges in understanding the mechanical behavior and micro-crack evolution patterns of deeply buried granite under different joint inclinations during pressure damage.

## 4. Conclusions

The specimens of the granite were subjected to uniaxial compression tests with different joint dip angles. The impact of joint angle on the mechanical properties and damage patterns of the specimens was thoroughly assessed using various criteria. Moreover, an analysis was performed on the acoustic emission characteristics associated with various joint inclinations. The following key findings are summarized from this study:

(1) The mechanical properties of granite specimens with prefabricated joints are lower than those of intact granite specimens. An increase in the joint angle α has been shown to be positively correlated with an increase in the specimen’s peak strength. Stress–strain curves can be used to categorize the rock damage process into three phases: initial compaction, crack expansion, and final damage. This process captures the rock’s complete mechanical behavior from initial loading to final damage.

(2) The spatial distribution of the acoustic emission events is consistent with the fine-scale damage patterns observed by X-ray computed tomography (X-CT). In the initial compression stage, primary micro-crack closure dominates damage evolution; the smaller the joint inclination angle, the more obvious the closure effect. In the crack extension stage, the fracture mode changes to mixed mode, eventually dominated by shear micro-cracks. Shear bifurcation extension dominates at α ≤ 45°, while directional main cracks form at α ≥ 60°. In the damage stage, the joint inclination angle significantly impacts the stress concentration area and crack type, altering the spatial distribution of the fracture network. An increase in joint inclination may accelerate the evolution of micro-crack fracture patterns, allowing shear cracks to dominate micro-cracks at an earlier stage.

(3) The inclination of joints has a significant effect on the paths of crack extension and damage modes. Joint inclinations of 0~45° promote stress concentration and inhibit transverse crack extension. In contrast, joint inclinations of 60~90° have a weaker effect on crack extension paths. All specimens exhibited brittle splitting-shear composite damage under uniaxial compression, with the main crack inclination angle decreasing as the joint inclination angle increased.

(4) The b-value algorithm was used to predict b-values that better correspond to the stress–strain curve, analyzing the scaling features of micro-fracture evolution in granite tests with varying joint inclinations. The b-value has a tendency to increase in a wave-like manner and decreases sharply near the peak of the compressive strength. During the initial loading stage, the b-value fluctuates upward, reflecting a high proportion of micro-cracks. As the test progresses, micro-cracks accumulate and form through-cracks, causing the b-value to decrease and signaling damage to the specimen.

## Figures and Tables

**Figure 1 sensors-25-03332-f001:**
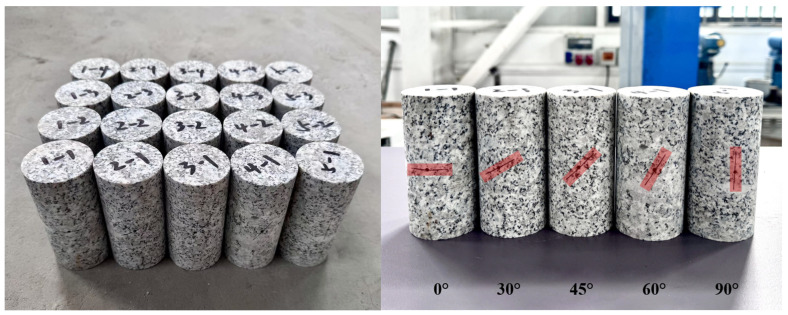
Prefabricated jointed granite specimens.

**Figure 2 sensors-25-03332-f002:**
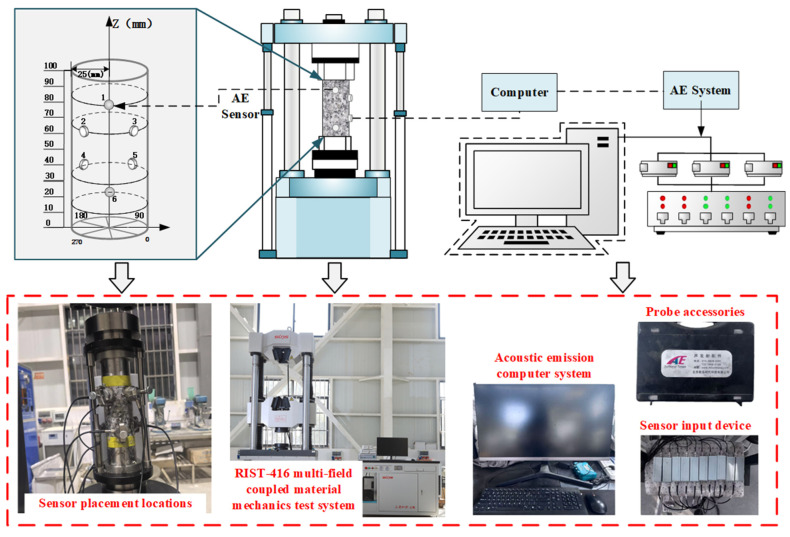
Flow chart of the test.

**Figure 3 sensors-25-03332-f003:**
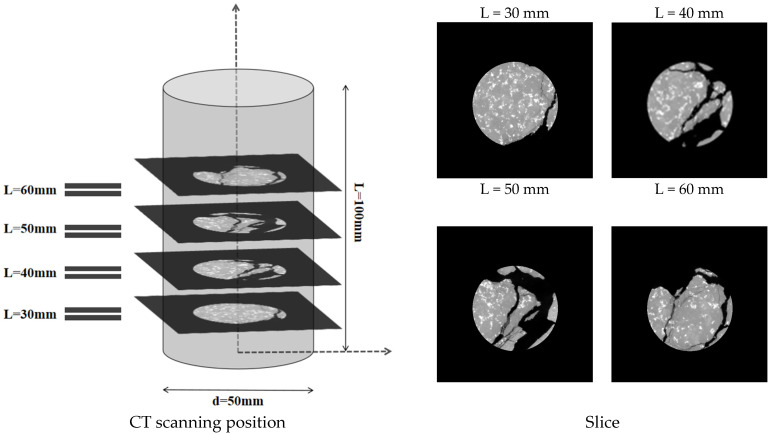
CT slice selection scheme (middle, upper, and lower scan positions selected for CT imaging).

**Figure 4 sensors-25-03332-f004:**
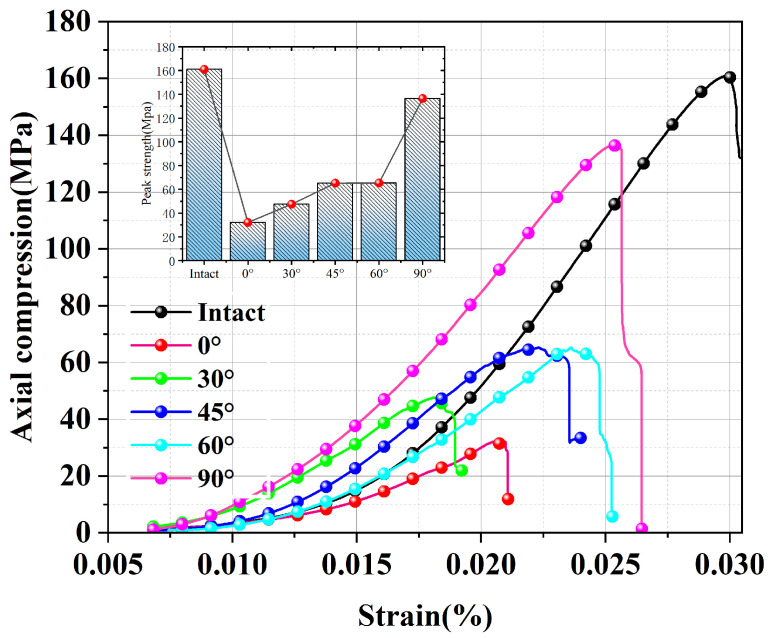
Stress–strain curves of granite specimens under uniaxial compression conditions.

**Figure 5 sensors-25-03332-f005:**
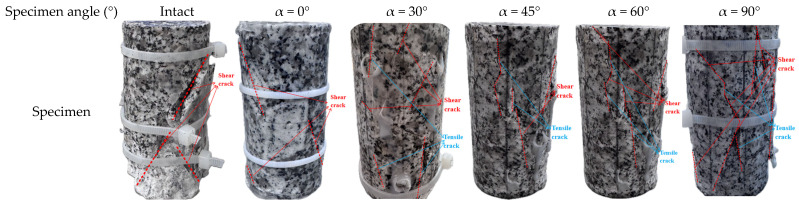
Post-failure images of different jointed granite specimens.

**Figure 6 sensors-25-03332-f006:**
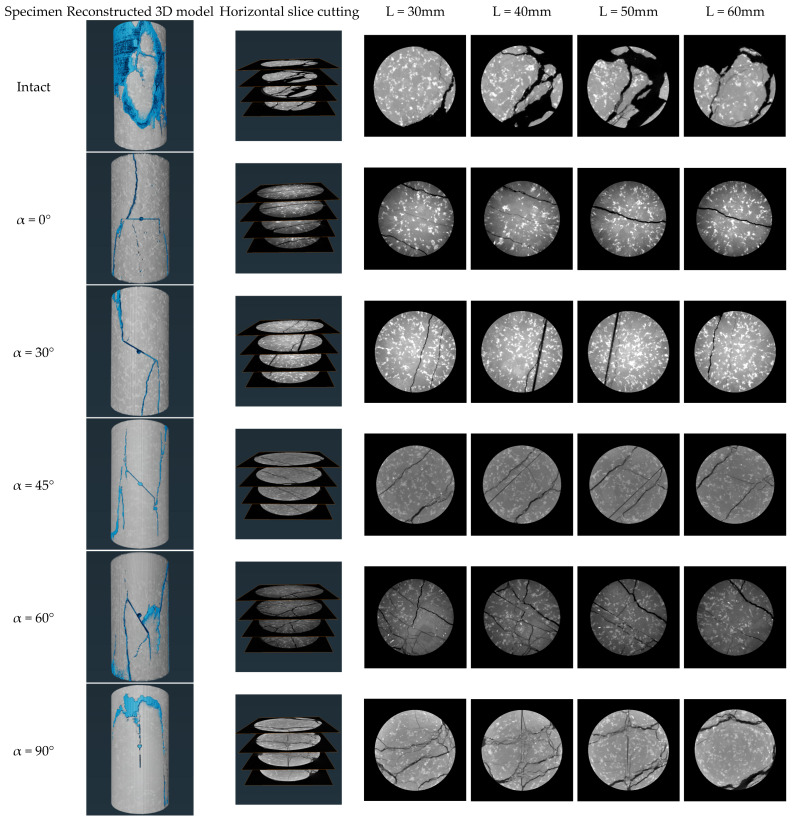
X-CT slices of internal fracture morphology of granite specimens after the test. (The blue area indicates the crack location).

**Figure 7 sensors-25-03332-f007:**
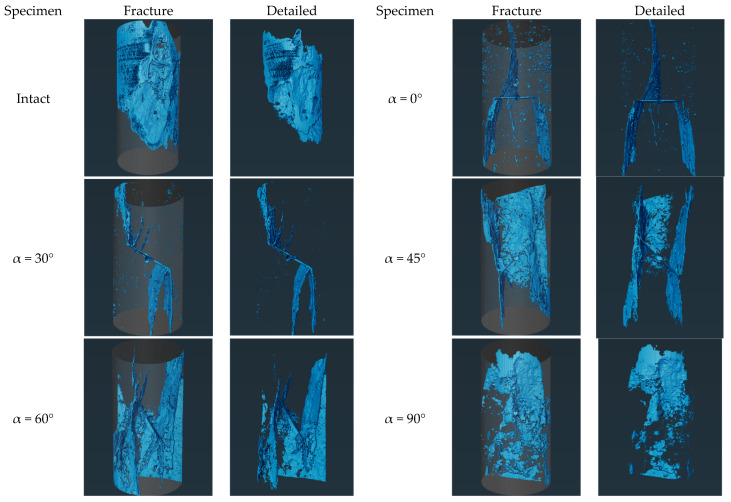
Extraction of cracks from reconstructed CT images of intact granite and granite specimens with different joint inclinations. (The blue area indicates the crack location).

**Figure 8 sensors-25-03332-f008:**
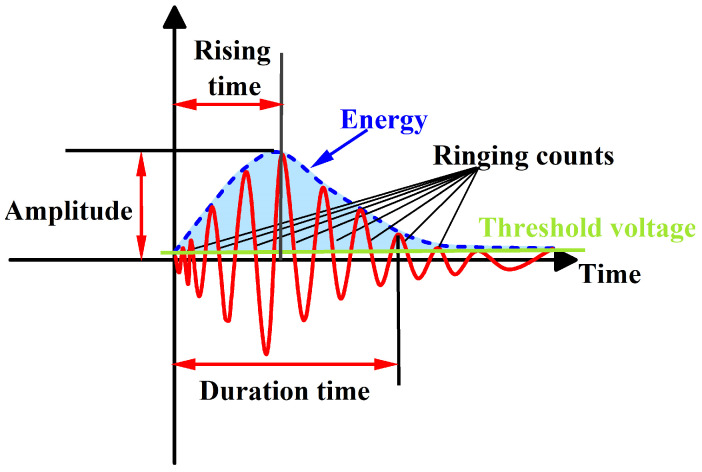
Parameters related to acoustic emission (AE) signal waveforms.

**Figure 9 sensors-25-03332-f009:**
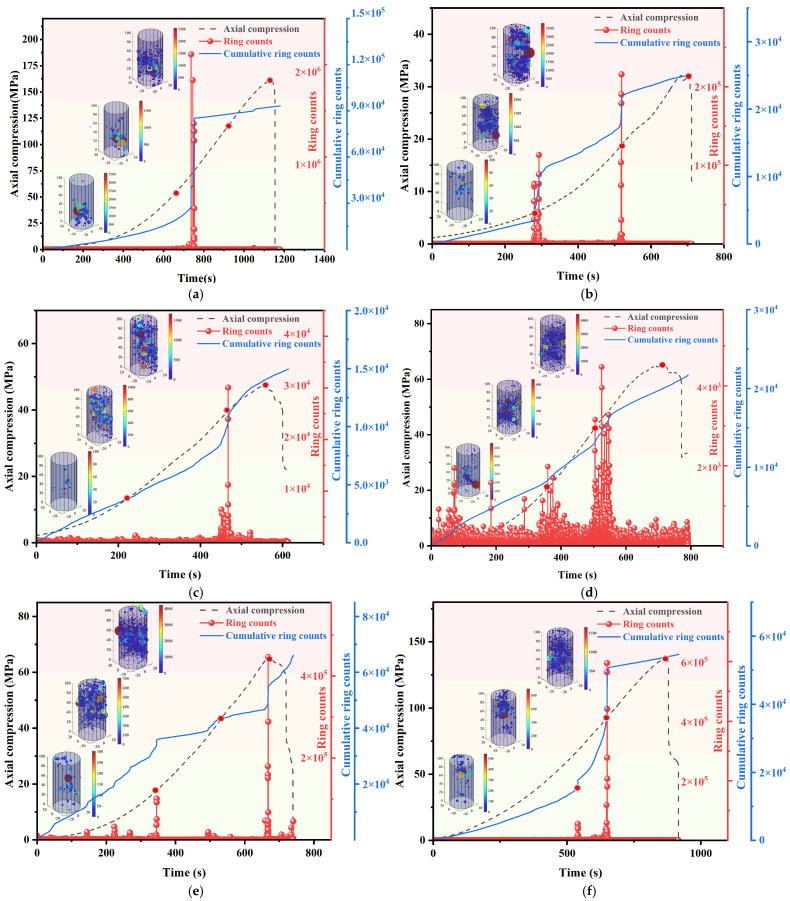
Variation in AE ringing counts and location point energy with stress and time, (**a**) intact (**b**) 0°, (**c**) 30° (**d**) 45°, (**e**) 60° (**f**) 90°.

**Figure 10 sensors-25-03332-f010:**
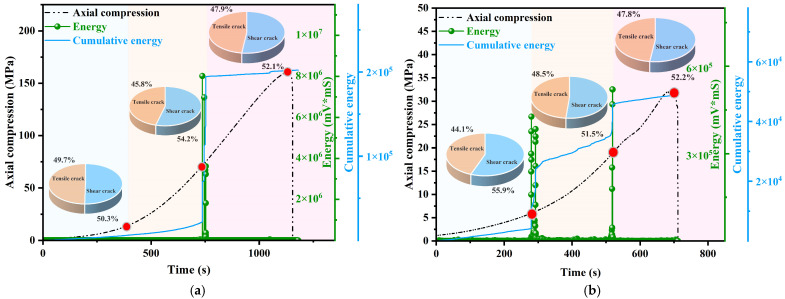
Acoustic emission energy and shear-tension crack evolution of granite specimens, (**a**) intact (**b**) 0°, (**c**) 30° (**d**) 45°, (**e**) 60° (**f**) 90°.

**Figure 11 sensors-25-03332-f011:**
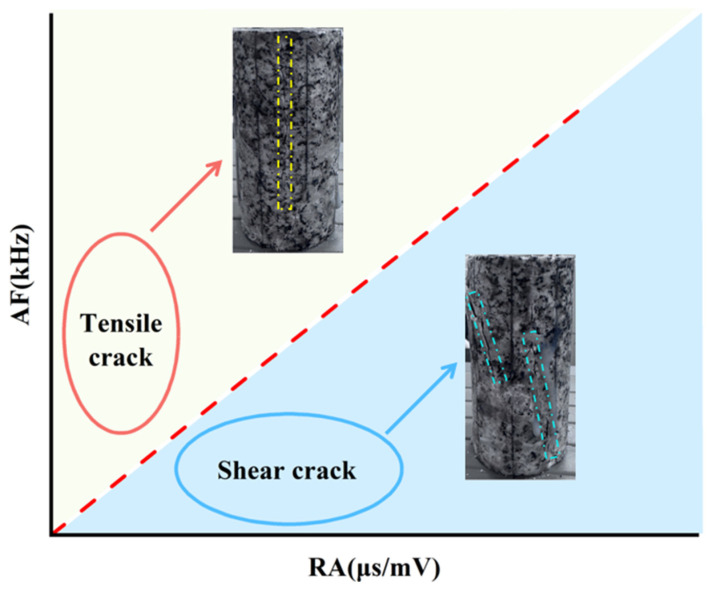
Schematic diagram of the method for classifying cracks in specimens based on RA and AF.

**Figure 12 sensors-25-03332-f012:**
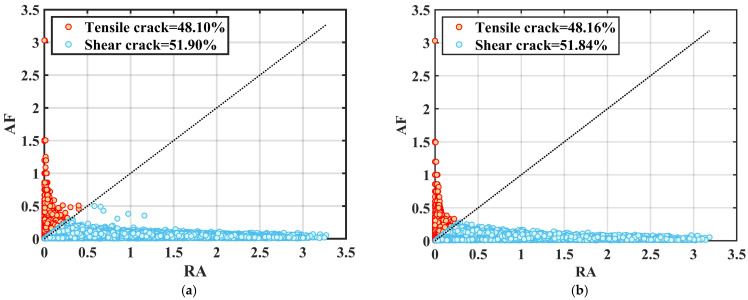
Characteristics of the RA-AF distribution of granite specimens at different joint inclinations, (**a**) intact (**b**) 0°, (**c**) 30° (**d**) 45°, (**e**) 60° (**f**) 90°.

**Figure 13 sensors-25-03332-f013:**
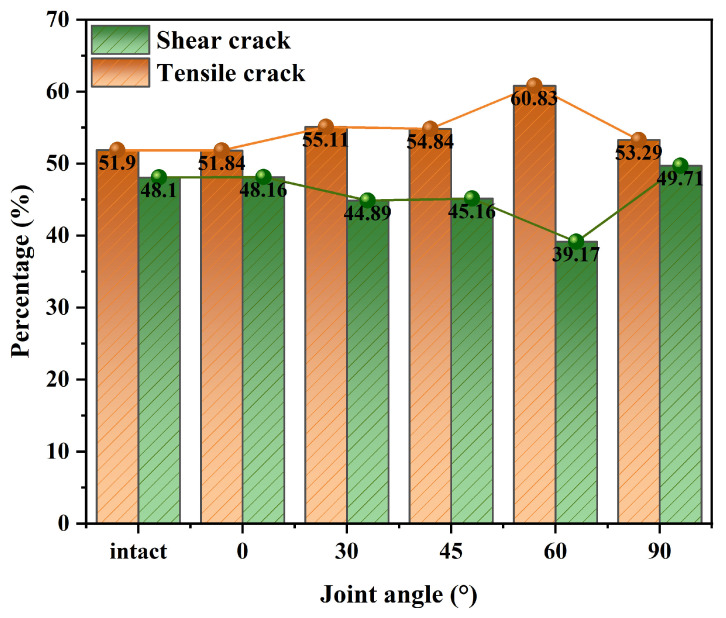
Variation in the tensile-shear crack ratio in granite specimens with different joint inclinations.

**Figure 14 sensors-25-03332-f014:**
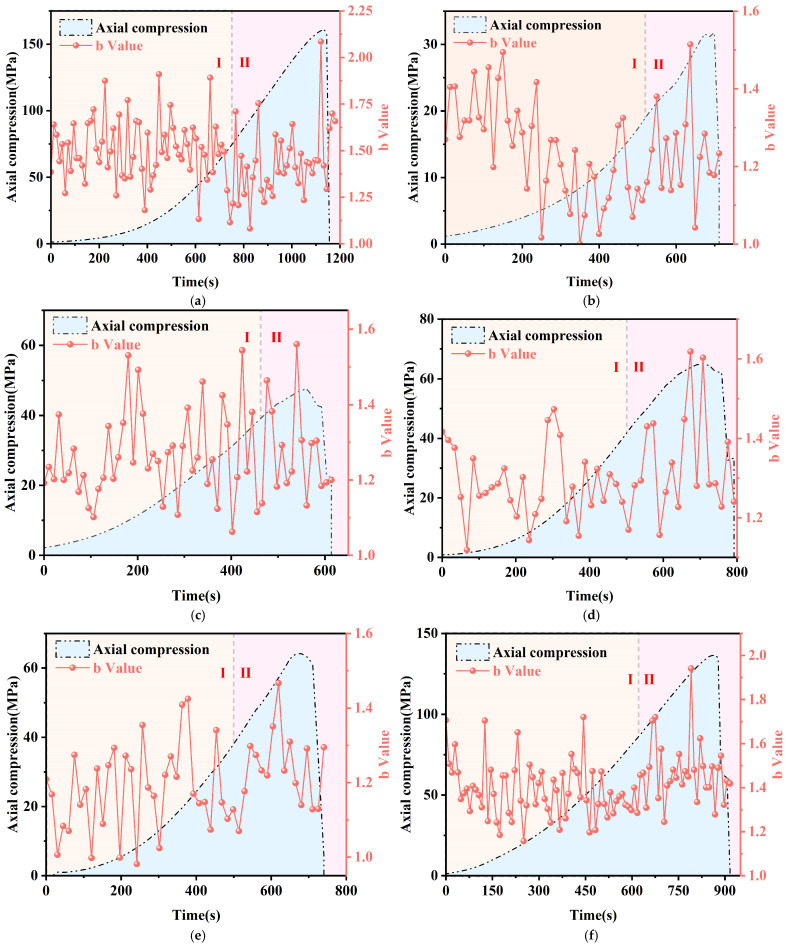
Changing law of acoustic emission of the b-value of granite specimen under uniaxial compression, (**a**) intact, (**b**) 0°, (**c**) 30°, (**d**) 45°, (**e**) 60°, (**f**) 90°. (In the figure, the markers I and II correspond to the Initial compaction stage and the Crack propagation stage, respectively).

**Figure 15 sensors-25-03332-f015:**
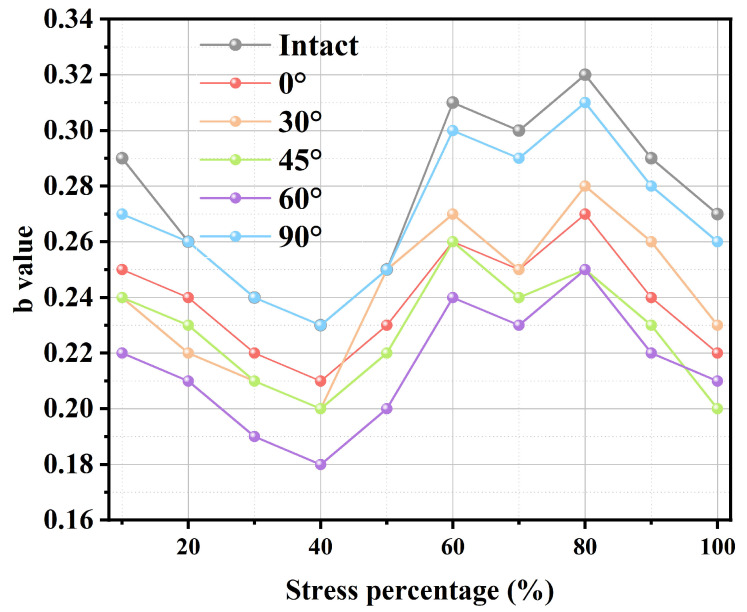
Variation curve of the b-value of granite specimens with different stress levels.

**Table 1 sensors-25-03332-t001:** Mechanical parameters of granite specimens.

Specimen Angle (°)	Diameter (mm)	Height (mm)	Weight (g)	Volume (m^−3^)	Density (kg/m^−3^)	Wave Velocity (m/s)	E (GPa)	Peak Intensity (MPa)	Breaking Time (s)	Post-Peak Destruction Time (s)
Intact	49.98	100.21	528.90	19.660	2690.17	3900	5.397	160.97	1153.87	45.03
0°	49.97	100.06	518.54	19.599	2645.66	3748	2.202	32.1	706.4784	7.1404
30°	49.96	100.28	519.30	19.658	2641.61	3869	3.590	47.5	561.9681	57.5017
45°	49.96	100.18	521.76	19.638	2656.77	3452	3.700	65.2	775.7877	83.4971
60°	49.95	100.16	517.87	19.627	2638.55	3156	5.003	65.3	840.7624	82.4925
90°	49.87	100.99	521.54	19.726	2643.87	3075	4.993	136.4	960.082	52.5097

## Data Availability

The data generated and analyzed for this study are available from the corresponding author upon reasonable request. Due to the contractual agreement with the participants, if there is a need for these data, they can be obtained by contacting the corresponding author.

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
