# Peer review of "Uncovering the Damage Mechanism of Different Prefabricated Joint Inclinations in Deeply Buried Granite: Monitoring the Damage Process by Acoustic Emission and Assessing the Micro-Evolution by X-Ray CT"

_sensors, 2025, doi:10.3390/s25113332_

Round 1

Reviewer 1 Report

Comments and Suggestions for Authors

This paper discusses the damage mechanism and fracture evolution characteristics of deeply buried granite with prefabricated joints under uniaxial compression monitored by AE technology. It is scientific sound with encouraging results. It deserves to be published. However, I still have the following comments for the authors.

  1. In Fig.3, it should be L=60 mm rather than d=60mm.
  2. In Fig. 6, why the X-CT slices morphology for the intact sample is smaller compared with the others?
  3. In Fig. 12, the tensile crack for the intact sample is 48.10% or 48.1%?

Reviewer 2 Report

Comments and Suggestions for Authors

The authors investigated the mechanical behavior and damage mechanisms of deeply buried granite with prefabricated joints under uniaxial compression, by using AE monitoring and X-CT imaging. The major findings include the correlation between joint angle and peak strength, shear and tensile cracks, and the utility of AE parameters. The work provides insights relevant to underground engineering in high-stress environments. Here I have some scientific questions:

  1. While the combination of some of the key words, e.g., deeply buried granite, prefabricated joints, AE monitoring, and X-CT, is not uncommon, the authors should emphasize what is novel about this work.
  2. As Fig.1 shows that five species exist for each joint angle. How consistent are the results with different species? Assessing the standard deviation may increase the confidence of the results.
  3. In the second panel from the left in Fig. 6, five X-CT image slices exist, but only four are explicitly shown. Why is that?

Besides the scientific questions, I list some other minor issues here:

  1. Affiliation 4 is listed but not associated with any of the authors.
  2. A few formatting issues:
  • On page 1, the parentheses in “(e.g., joints, fractures, and pores” are not closed
  • Several instances of 'AE' and '(AE)' on page 2 seem to lack meaning and may be editorial mistakes.

     3. Grammar issues:

  • On page 11, in “Regarding intact granite specimen parts, the bottom of the specimen is very little in the acoustic emission events, …”, “very little” should be “very few”.
  • On page 11, “ the specimen's ringing counts of the specimen increases in stepwise” reads awkward.

     4. In Fig.10, the legend has a typo of “axial compression” as “aixal compression”; also, the legend of it should be a dashed line rather than a dashed box, as in Fig.9. 

This list is by no means complete. Please proofread the entire manuscript carefully.

Reviewer 3 Report

Comments and Suggestions for Authors

This research reveals the damage mechanisms and fracture evolution characteristics of deeply buried granite prefabricated joints. which is very interesting and valuable for engineering practices. This paper can be accepted for this Journal after a minor revision.

My comments are listed here,

(1) In Figures 12 and 13 of the article, the description of the pattern of change in the percentage of tensile and shear cracks in granite with different inclinations is too quantitative, so please add some descriptions qualitatively.

(2) Please check the grammar of the article in detail to avoid some unprofessional descriptions, such as the expression ‘(AE) a useful way’ in line 66 of the manuscript.

(3)Conclusion 2 merely describes the application of X-CT technology in uniaxial compression testing and does not form a quantitative conclusion and is recommended for deletion.

(4)The conclusion of the paper is too redundant, please streamline the conclusion and highlight the innovations.

Round 2

Reviewer 2 Report

Comments and Suggestions for Authors

The authors has addressed the comments, and I recommend the publication of this work.